# Drone-Based Gamma Radiation Dose Distribution Survey with a Discrete Measurement Point Procedure

**DOI:** 10.3390/s21144930

**Published:** 2021-07-20

**Authors:** Andras Molnar, Zsolt Domozi, Istvan Lovas

**Affiliations:** 1John von Neumann Faculty of Informatics, Obuda University, Becsi ut 96/b, H-1034 Budapest, Hungary; molnar.andras@uni-obuda.hu; 2Doctoral School of Applied Informatics and Applied Mathematics, Obuda University, Becsi ut 96/b, H-1034 Budapest, Hungary; lovas.istvan@uni-obuda.hu

**Keywords:** dose distribution measurement on field, radioactive mapping with drone, radiation measurement with drone

## Abstract

A dose distribution map can be created using geographic information system (GIS) methods from sensor data that do not provide image information in a classical way. The results of discrete radiation measurements can be properly represented in a uniform raster above the surface. If the radiation measured at each site does not show a jump-like change, a dose distribution map can be prepared by interpolating the measured values. The coordinates of the measuring points can be used to calibrate the map. The calibrated and georeferenced map is suitable for locating hidden or lost radiation sources or for mapping active debris scattered during a possible reactor accident. The advantage of the developed method is the measurement can be performed with a small multicopter, cost-effectively, even without human intervention. The flight time of small multicopters is very limited, so it is especially important to increase the efficiency of the measurement. During the experiments, a practical comparison of several methods was made with regard to the measurement procedure. Similarly, based on the measurement experience, the detector system was further developed and tested in three main steps. A system was developed with a detector system with a total weight of 500 g, including a battery capable of operating the detector for at least 120 min. The device is capable of detecting an average of 30 events/min at of 0.01 μSv/h background radiation. Experiments have shown that the system is able to significantly detect a source with an activity of 300 μSv/h by scanning above 10 m ground level.

## 1. Introduction

Several dose distribution maps have been created using a drone-mounted gamma radiation detector. Based on the results and the experiences gained in the experiments, the shortcomings of the system, as well as the possibility for further development have also emerged. The primary objective of the development was to create a more compact, easily portable, and deployable system, but one which, in contrast with the previous ones, is more sensitive. Despite background radiation (typically 0.01 μSv/h in the testing areas), a difference of +0.009 μSv/h has reliably been detected in the experiments made so far. In this present development, this value was successfully reduced to +0.005–+0.007 μS/h. The improvement in sensitivity was achieved primarily by increasing the measurement time per point, which was realized using special flight control software. This sensitivity increase either allows higher scanning altitude (approximately +1–2 m) or, in case of an identical flight device at an identical altitude, a larger survey area with one take-off. Naturally, scanning height or scanning speed can increase significantly if the activity of the searched source is high. In our experiments, we used a natural uranium mineral (Autunit) with activity far below that of artificially produced isotopes. In this series of experiments, we also covered the identification of several sources, which models the possibility of mapping active sources scattered around the site of a possible accident.

The main advantage of the system developed and introduced by us over the survey procedures used in practice is that it is easy to mobilize, a large area can be surveyed at low cost without putting an operator at risk in the field. The purpose of the system is to detect the presence of the source and to localize it to such an extent that the localization can subsequently be easily specified by manual or other ground procedures. Since we do not strive for centimeter positioning accuracy, standard GPS localization is sufficient for measurements. During the measurements, the geographical coordinates are interpreted in the WGS’84 system. The circles of latitude and longitude coordinates are also shown in this system in the figures presented.

## 2. Applied Detectors

From radiation-sensitive detectors that can be mounted to a drone, two were chosen for testing [1]. Since defining radiation energy was not a goal, the Geiger-Müller tube particle counter is a convenient detector that can easily be adapted to digital devices. It is an obvious choice for testing this detector in drone surveys. However, our previous experiments on the subject [2,3] have also demonstrated the disadvantages of these detectors. To be precise, this is primarily the relatively large mass of the constructed system. Modern semiconductor-based scintillation detectors can be promising alternatives to performing the given task. Several projects use such detectors [4,5,6,7]. Therefore, we also examined the use of such a detector in a specific measurement. In the case of the scintillation detector, we did not test the energy of the detected particle, in this way we treated this detector only as a particle counter.

### 2.1. Geiger-Müller Detector

The Geiger-Müller counter is an electric discharge tube filled with low-pressure gas capable of detecting ionizing radiation. In general, it can be sensitized to specific radiation with the material of the tube or its coating or with its end-window design. During operation, the device provides electric impulses for all detected photons. That is why the Geiger-Müller detector is often called a particle counter. Although the tube output signal is analogue, the information content is in impulses that can be counted over a unit of time (not in the impulse amplitude!), therefore it can be connected to digital systems relatively easily without using an analog/digital (A/D) converter. It has a simple design. It consists of a metal cylinder filled with low-pressure gas. In the axis of the cylinder, there is a thin electrode isolated from the cylinder. To operate, the pipe (cylindrical surface and central electrode) must be connected to a direct current (DC) with the correct polarity. This voltage is around 500 to 800 V depending on the type of pipe. Each impulse means an electric discharge inside the tube. During discharge, the tube is not capable of detecting another photon. This time is called dead time. The dead time of Geiger-Müller tubes depends on the type but generally falls in the order of 10^−5^ × 10^−5^ s. The effect of dead time should be taken into consideration during the measurement. This effect is not significant in the case of weak radiation, but as radiation increases, the probability of undetected photons increases. In principle, the impact of the photons into the detector is random. Therefore, even at low radiation, during the dead time caused by a photon being detected, one or more photons may be absorbed in the detector, but they do not cause another discharge, i.e., an electrical impulse. The electric discharge in the Geiger-Müller tube causes ionizing radiation but based on the discharge, i.e., on the resulting electrical impulse, the energy of the detected photon can not be deduced. It follows that the nature of radiation (the energy of gamma photons, such as the isotope causing decomposition) cannot be inferred using Geiger-Müller tubes. At the same time, the Geiger-Müller tube is a very simple and well-functioning detector that can easily be connected to processing units (either to a speaker that indicates the intensity of radiation with click frequency or to a counter that displays the detected particle count per a given unit of time), thus as a simple radiation detector, it is rather widespread.

### 2.2. Scintillation Detector

Certain crystals emit light when exposed to gamma radiation. In fact, the emission of light is not continuous, but each gamma photon absorbed in the crystal produces a flash of light proportional to its energy. This is the phenomenon of scintillation. If the crystal is attached to a very sensitive light sensor and the light sensor is isolated from all other light sources, flashes can be detected in the form of electrical impulses. The PMT (photo multiplier tube) is the first such light sensor, which can still be used today. A typical scintillation detector thus consists of two main elements. One is the scintillation crystal itself, the other one is the PMT optically coupled to the crystal. A great advantage of such detectors is that the density of the crystal is significantly higher compared to ionization detectors so that a sensitive device can be built in a smaller size. In addition, scintillation detectors provide a signal proportional to the energy of the detected gamma photon, which allows the determination of additional characteristics of the source under study. The disadvantage of conventional scintillation detectors is precisely the use of PMT. It is a relatively large, high-voltage device that is sensitive to external electromagnetic noise.

Semiconductor-based light sensors were developed to serve to replace PMT in modern devices. One of these is the MPPC (multi-pixel photon counter). These have several advantages over PMT. They can be operated at low voltage, have low power consumption, are less sensitive to electromagnetic fields, and are significantly smaller devices. Naturally, a stabilized power supply and heat compensation must be provided during their use.

Scintillation crystals of different materials are used depending on the magnitude, energy, and intensity of the radiation. The most common crystal is Thallium-contaminated (activated) sodium iodide NaI (Tl) [8]. Apart from the many advantageous properties of this crystal, its water-binding capacity is particularly significant, which is a great disadvantage. The optical property of the crystal deteriorates due to the bound water (it turns yellow, becomes opaque), which eventually means a significant weakening in the efficiency of the detector. Therefore, the NaI (Tl) crystals in the detector must be hermetically sealed from the outside environment so that they are not damaged by air humidity.

The combination of the crystal and the light sensor must be in harmony. The wavelength of the light emitted by the scintillation crystal and the detection wavelength of the light sensor must be taken into consideration. For example, CsI (Tl) (Thallium-activated cesium iodide) can be fitted to MPPC sensors well [9,10,11]. However, the CsI crystal is soft, fragile, and absorbent. Generally, this crystal is provided with an outer coating primarily to reduce surface vulnerability.

## 3. Measurement Procedure

The measurement procedure can be divided into two phases. In the first phase, on the planned route, the device hovers at the designated measurement points (which form a raster grid) for a predetermined amount of time and then flies to the next measurement point. Hovering time was determined to be 10 s. Thus, corresponding to the sensitivity of the detector, we can record several pieces of data at each point whose coordinates are constant. More data collected at the same point provide a better signal-to-noise ratio (radiating sample/background radiation), which increases the probability of detecting a detectable source. If the operating time of the carrier does not allow the survey of the test area, this may be managed from several take-offs.

For this phase of the survey, we developed a flight control software solution. Unfortunately, the flight control software available for DJI aircraft fundamentally supports aerial photography, thus hovering the drone over a position for a long time is not possible. The software we developed runs under the Android operating system and was created using the SDK provided by DJI. The software is essentially a route planner, which allows hovering at user-set heights for evenly spaced waiting points along the route. Thus, it was possible to reliably repeat the surveys even at extremely low altitudes (2–3 m above ground level). The hovering time assigned to the waiting points can also be set by the user. The software checks the planned flight program and compares it with the maximum flight time with the battery capacity of the given DJI aircraft. If the plan involves a flight longer than what the drone is capable of, it will not allow takeoff. In this case, several flight plans must be prepared which cover smaller sections and they must be flown one after the other.

In the second stage of the measurement, the data recorded during the flights are processed offline, using a self-developed MATLAB software, and the result is then rendered.

As the first step of data processing, the measurement area is divided into the same sub-areas as shown in Figure 1.

A bounding rectangle-shaped area must be $\Omega_{i,j}\subset R^{2}\;\mathrm{spanning}\;\mathrm{from}\;X_{min}\;\mathrm{to}\;X_{max}\;\mathrm{and}\;\mathrm{from}\;Y_{min}\;\mathrm{to}\;Y_{max}$ where
(1)$$X_{min}=\underset{p\in P}{\min}p_{x}$$
(2)$$X_{max}=\underset{p\in P}{\max}p_{x}$$
as well as
(3)$$Y_{min}=\underset{p\in P}{\min}p_{y}$$
(4)$$Y_{max}=\underset{p\in P}{\max}p_{y}$$
where p is the GPS coordinate of a measurement point (*p_x_*, *p_y_* ∈ R), furthermore p is the set containing all points. The bounding rectangle with n × m divisions is formed as follows:
(5)$$\begin{array}{cc} \Omega_{ij}= & \left\{\begin{array}{cccc} \left[x^{i},x^{i+1}\right)\;\times & \left[y^{j},y^{j+1}\right) & if & \;i\in \left[0,n,\ldots ,n-2\right]\;and\;j\in \left[0,m,\ldots ,m-2\right]\\ \left[x^{i},x^{i+1}\right]\;\times & \left[y^{j},y^{j+1}\right) & if & \;i=n-1\;and\;j\in \left[0,m,\ldots ,m-2\right]\;\\ \left[x^{i},x^{i+1}\right)\;\times & \left[y^{j},y^{j+1}\right] & if & \;i\in \left[0,n,\ldots ,n-2\right]\;and\;j=m-1\;\\ \left[x^{i},x^{i+1}\right]\;\times & \left[y^{j},y^{j+1}\right] & if & \;i=n-1\;and\;j=m-1\; \end{array}\right. \end{array}$$
where
(6)$$x^{i}=x_{min}+i\cdot dx$$
and
(7)$$y^{i}=y_{min}+i\cdot dy$$

Based on the above, we determine an $f\left(p\right)$ function that gives the value of  p points. The value for the cell at (i,j)- can be determined using the following relation:(8)p¯Ωi,j=∑ fpp∈Ωi,j p:p∈Ωi,j

The averaging function can be replaced by any other function, e.g., median. Thus, the measurement area is divided based on predefined parameters. Each measurement point will belong to a specific cell. The value of a cell, depending on the settings, is given by the average, median, or maximum of the measured gamma rays.

## 4. Practical Experiences

To test the method in practice, two drones with distinct detector systems were used. Measurements were performed sequentially under the same conditions for the results to be comparable. During the planning phase of the experiment, we took into consideration our experience in measuring gamma radiation with continuous flights [1,2].

### 4.1. Geiger-Müller Detector-Based Measurements Mounted on a DJI Inspire Drone

Inspire is an older product from DJI. Its large size and load capacity make the drone suitable for carrying larger payloads of up to 1000 g. Its reliable flight capability is particularly advantageous for the delivery of the Geiger-Müller detector system assembled for radiation measurement. The detector system itself consists of four high-sensitivity GM tubes (LND 7808) [12], of power supplies supplying the suitable voltage, a GPS module, and a microcomputer recording the measured data. The system is designed so that the measured data of each detector are stored separately at unit intervals, as well as the combined measured data of the four detectors that are stored too. Consequently, the measurement efficiency of the duplication of the detector tubes can be examined afterwards.

The flight time of the drone with the detectors fitted is 10 to 18 min. This amount of flight time was not sufficient to survey the 27-m-long and 24-m-wide testing area. The area was divided into a 3 × 3 m square grid. At each of the grid points, the drone hovered for 10 s. The measurement was performed with six takeoffs. During the landings, the GPS unit of the detector system operated continuously. We tried to minimize the errors of the measuring points in this manner. During the six flights, the area was scanned twice for the measurement grid points to eventually be 1.5 m apart.

The gamma radiation source located in the testing area consisted of Autunite (Ca(UO_2_)_2_(PO_4_)_2_ × (10~12)H_2_O) minerals spread on a disk with a 30 cm diameter. This mineral is a natural occurrence of Uranium. The activity of the minerals used in the experiment was below the threshold limit value. Its acquisition and possession in small quantities are not bound to a license. Figure 2 illustrates the gamma radiation spectrum of an Autunite sample. The spectrum was recorded with a Scintillation detector. The recording was evaluated using a program available for free download called “BecquerelMonitor” [13]. During the recording of the spectrum, the sample was in the immediate vicinity of the detector. The distance between the scintillation crystal and the sample was 3 mm due to the structural design of the detector. Recording time took 60 min. The figure clearly shows the increased radiation compared to the background radiation recorded as a control [14,15]. Geiger-Müller detectors are not suitable for measuring the energy level of detected gamma particles, only their presence can be detected. However, it is clear from the diagram that the Autunite sample generates a definite increase in gamma particle number. Based on this fact, we used an Autunite mineral as a radiation source in further experiments.

For the experiment, we had to determine the measurement distance (flight altitude) at which the source used can still be expected to be detected. A simple geometry (Figure 3) was used to model the spread of radiation with the following hypotheses:the source is considered to be a point from the measuring distance,the weakening of gamma radiation in light of the distance traveled in the air is negligible in the measurement range, i.e.,
(9)$$I=I_{0}e^{-\mu r}≅I_{0}$$
where*I*: radiation calculated at distance *r*,*I*_0_: radiation measured at the reference point,*μ*: coefficient of the weakening of the medium,*r*: distance from the source (reference point).the source is purely a gamma emitter at the measuring distance,the radiation is isotropic.


Figure 4 illustrates the results of the experiment which was measured at different distances with the number of samples used in subsequent studies. The purpose of the static measurement was for the measurement results to become comparable with the theoretical calculations and to determine the minimum and maximum measurement distances. It can be read from the figure that the measurements advisable to be performed at a height of no more than 6 m from the sample because from this height the source signal can no longer be separated from the background radiation with a short measurement time. Although measurements made lower than 3 m give detectable results, the initial restrictions are no longer fully met. As the source used in testing consists of several tiny minerals, the combination of these can no longer be considered a point at close range. Consequently, the measurement will be burdened with major errors which will not allow quantitative evaluation.

During the experiment, 1620 measurements were performed at nearly 162 measurement points. The measurement data are illustrated in the diagram in Figure 5. It can be clearly seen that the detectors detected only slightly different (0.015–0.017 μSv/h) activity from the typical background radiation measured in the area (0.01 μSv/h) during the measurements. On the diagram, the high value of the gray line shows the flight phase, while the low value shows the time spent on the ground (battery replacement). Among the bottom lines of the chart, the high value of the thin, continuous line shows the peaks generated by the sample, while the high value of the dashed line most likely shows the data belonging to the sample. As the experiment was designed so that the physical location of each measurement point and the sample should not show a relationship relative to each other, measurement over the sample was not guaranteed. However, the densely placed measurement points guaranteed that there will be several measurements that are close enough to the sample, in this way it will already change the measured radiation value. This results in that there will be measurement points that indicate the effect of the sample correctly and there will be measurement points slightly differing from background radiation, but which can also be connected to the sample. Although these measurement points are included in the diagram, they cannot be clearly separated without further information.

In Figure 5, the detected radiation data were filtered with a five-element median filter. The reason for this is that the measurements were taken at discrete points. At each point, the system recorded 10 measurements in 10 s. The median filter filtered out the peaks that were recorded for the duration of only one measurement. Due to the natural character of radiation, these short impulses may occur, but do not indicate the presence of persistently high radiation. We presumed that the sample would generate persistently elevated values through multiple measurements instead of short-term impulses.

Figure 6 illustrates the unfiltered data from the measurement. As visible, it is difficult to find the sign of the sample in the unfiltered data set, and several false peaks also appear.

Given that the coordinate of the measurement is also recorded for each radiation measurement, this additional information can be used in data analysis [16,17,18]. The test area was divided into m × m components. During the analyses, the value of “m” was 1; 2; 3, and 4 m. This division size is adjusted to the distance of discrete measurement points determined during flights. The radiation data were assigned to each sub-area and these sub-areas were described by the median of the radiation values assigned to them.

Figure 7 illustrates the planar gamma intensity distribution of the measurement data above [19,20]. The sample placed in the test area is distinctly separate from the rest of the area. The significant effect of the sample on the image is remarkable. The effect of the sample is significantly less distinct when observed on a diagram.

In order to check the accuracy of the geographical coordinate of the source indicated by processing, a control survey was performed. During the control survey, the drone hovered with the device exclusively over the sample in such a way that the detector was 50 cm from the source. As a result, we obtained values that were completely distinct from background radiation. A significantly higher value compared to the background radiation could only be measured at the coordinate above the sample. There were no other values in the area close to the radiation value measured at the sample. Thus, disturbing peaks were not generated during processing.

The recorded coordinate values were evaluated by statistical methods. In doing so, the outliers were omitted, and the median of the remaining values was taken as the static coordinate of the sample. Given that localization should primarily support the facilitation of the initial steps of manual search, measurement errors were within the expectations of localization.

Figure 8 illustrates the processing of control measurement data. It can be seen that only the location of the radiation source is outlined in the image. Arrows in the figure indicate the geographical coordinates of the source, which are identical to the location of the source detected during the processing of the test measurement shown in Figure 7.

### 4.2. Measurements Made Using a Scintillation Detector Fitted to a DJI Mavic Drone

DJI’s Mavic-type drones are relatively new developments. The drone itself is significantly smaller than the Inspire series. Its flight stability and accuracy also surpass older versions. Due to its smaller size, it is easier to transport, store and operate than its older, larger counterparts. However, its compact design makes it rather difficult to mount the accessories on the device. Its load capacity is also lower than that of the Inspire drones, thus it is not possible to install the already developed Geiger-Müller detector system on the Mavic drone. However, radiation meters with scintillation crystals possessing small and lightweight semiconductor sensors can be placed on the device. Thus, a small scintillation detector and a Raspberry PI, for data processing, were placed on the Mavic Pro drone. In the experiments, we examined how reliable the data provided by a scintillation detector are in flight; the detector had also already shown adequate sensitivity during static measurements. In terms of the structure of the detector used, it consists of a CsI (Tl) crystal with 13 × 13 × 47 mm cubature and an MPPC sensor. The detector detects an average of 30 particle hits with 0.01 μSv/h radiation (background radiation measured in the experimental area).

Given that our goal was to implement a small, easy-to-transport system, one of the carrier systems was a DJI Mavic Pro multi-copter. Due to the limited load capacity, the required minimum has been determined for onboard systems. Of course, in the case of RTK localization, determining the measurement points would provide an accuracy of less than a centimeter, but we did not aim for this accuracy in our system. According to our research and development concept, the measurement accuracy of the standard GPS (for civilian use) was sufficient.

As Mavic can fly 10 to 12 min with the detector system attached, the area was possible to be measured with two takeoffs. The drone measured for 10 s at each of the 36 measurement points at the first take-off and at 45 measurement points at the second take-off. The measurement results, as well as the coordinates of the measurement points were recorded by the Raspberry PI on an SD card.

Although scintillation detectors are suitable for determining the energy of the detected gamma particle, this was not taken into consideration in the present experiment. The recorded data contain only the number of detected gamma particles per unit time (Figure 9).

No significant spikes were observed in the measurement data shown in Figure 9. Based on the data series, it is not possible to say whether there was a radiation source with an intensity differing from the background radiation in the measurement area.

However, with the planar representation of the data (Figure 10) the source becomes visible. Figure 10 shows two sources (apart from the noises shown in the image, which is due to the reason that the intensity of the source at the measuring sites was slightly higher than the intensity of the background radiation). The diagram shown in Figure 10 also shows the GPS accuracy recorded during the measurement. It can be seen that in the second third of the measurement time the accuracy deteriorated (the value became higher). At the same time, the number of satellites used for the measurement reduced from 10 to 9 based on the data (this value is not displayed). Due to the increased measurement inaccuracy, the recorded GPS coordinates also differed from the more accurate measurement period.

Thus, the appearance of the two sources in Figure 10 can be explained by the increase in GPS error that occurred during the measurement. The distance between the two source points marked with a black arrow in the figure is ~4 m. This distance corresponds to a position error due to an increase in the measurement uncertainty.

### 4.3. DJI Mavic Drone-Based Dual-Source Survey

In further tests, we examined the possibility of detecting sources located far apart in the test area. The samples were 25 m apart and had almost the same level of activity. The experiment took place in a slightly larger area than before, for the placed samples not to fall on the borders of the measurement area. The testing area thus became 39 m long and 24 m wide compared to the previous measurements (27 m × 24 m). The number of measurement points was 117, divided into 9 rows and 13 measurement points per row. At each point, the device hovered for 10 s. This measurement could be accomplished with four takeoffs. The measurement results were evaluated as already described. The raw measurement data are illustrated in the diagrams in Figure 11.

Figure 11 shows that the background radiation resulted in a counts per second (CPS) value between 20 and 40. Near the sample, this value increased significantly. At the second flight, two peaks are visible. Both peaks belong to the same source. The two peaks occurred because in two of the measurement points the device was within detection distance. At the third flight, a relatively large peak can be seen, which was formed during the measurement near the second source.

The measured results are illustrated in Figure 12 in the form of a spatial distribution map by the method already described. In the picture, the locations of the two sources are clearly distinct, their coordinates correspond to their real position. Thus, in the examined area, with the method, the detection of several independent sources can be reliably accomplished. It should be noted, however, that the experimental system does not use an RTK positioning device, therefore the minimum separation distance for independent sources is 4 to 5 m depending on GPS errors that may occur during the measurement (see Figure 10 and its explanation).

Figure 13 is a 3D representation of the dual-source measurement already presented. Surface height is proportional to the measured gamma radiation, more precisely to the number of events per second (count per second, CPS) recorded by the detector. For the sake of better transparency, we have removed events within the range of the background radiation during the representation, thus the protrusions of the surface illustrate the excess radiation. Accordingly, the CPS values for that area can be read on the “Z” axis of Figure 13. Although the coloring is proportional to the CPS value, it has no absolute value. The color palette was adjusted on each graph for the best visual experience.

## 5. Conclusions

The two tests to compare the detectors were performed in one day, leaving the sample in an identical position with the same measurement parameters, in this way they can be subjects for objective comparison. It is visible that the identification of the sample was realized in both cases, however, there are apparent qualitative differences in the results. In the case of a survey using the Mavic equipped with a scintillation detector resulted in a noisier result due to the fact that the detector system was less sensitive. One reason for this is that the four-tube Geiger-Müller counter detector system has a significantly larger surface area than the 8 cm^3^ scintillation crystal. Although scintillation crystals interact with more gamma particles in the same volume as Geiger-Müller tubes due to their higher material density, still the probability of detection is slightly lower due to their small size.

In terms of being a carrier, the Mavic drone was able to position itself to the designated measurement points much more precisely and to hold its position there for the duration of the measurement. Its portability and small size make it a more manageable, more compact device compared to the robust Inspire drone.

The measurement at discrete points gave a more accurate result than continuous flight measurement because there was no need to correct the position coordinate resulting from the motion. Another advantage of measuring at discrete points is that detection time increased, which allowed better separation for sample signals from the background noise.

The examination of several sources has well-illustrated that groups of radioactive debris scattered in an area or larger separate debris can be detected and localized correctly by using the measurement method.

## Figures and Tables

**Figure 1 sensors-21-04930-f001:**
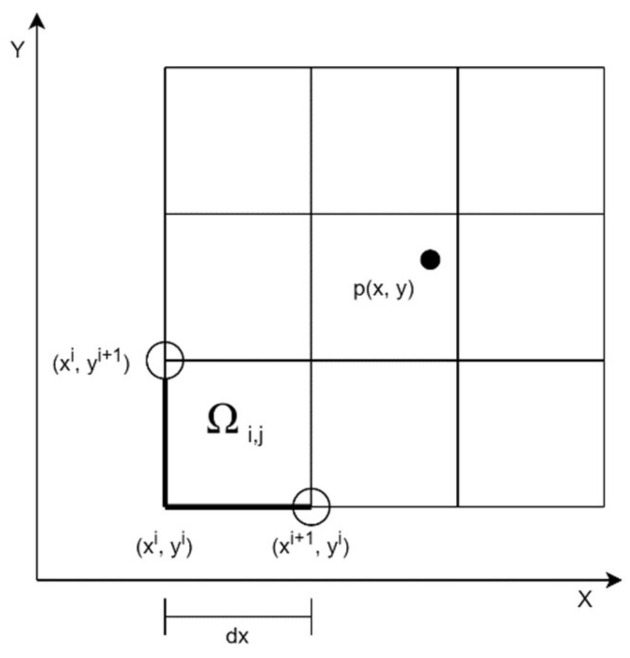
Dividing the area into cells.

**Figure 2 sensors-21-04930-f002:**
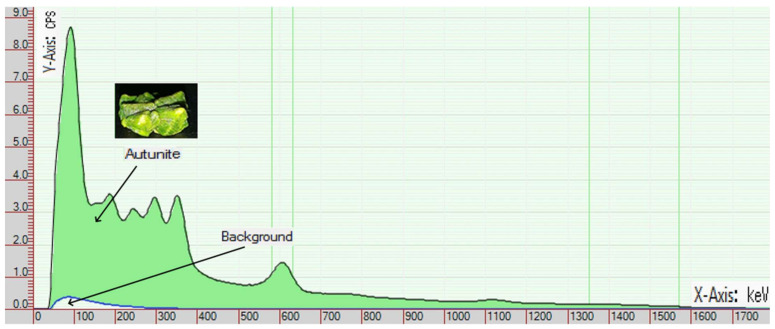
Shows the radiation spectrum of one of the Autunite minerals used in the experiments (Counts per second (CPS), kiloelectron volt (keV)).

**Figure 3 sensors-21-04930-f003:**
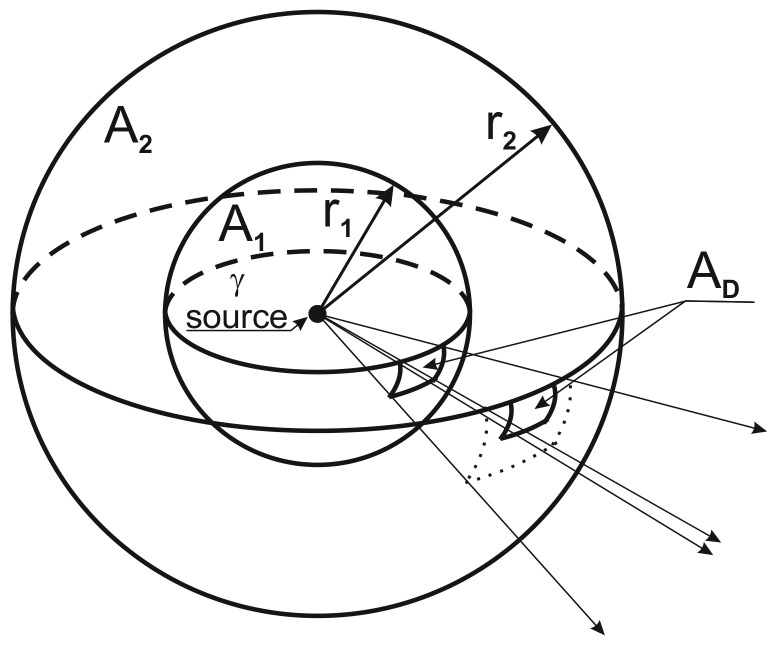
Modeling the distance-dependent detectability of the radiation. A_D_: the surface of the detector; A_1_: the surface of a sphere defining a unit distance; A_2_: the surface of the sphere determining the detection distance; r_1_: unit distance from the source; r_2_: detection distance from the source.

**Figure 4 sensors-21-04930-f004:**
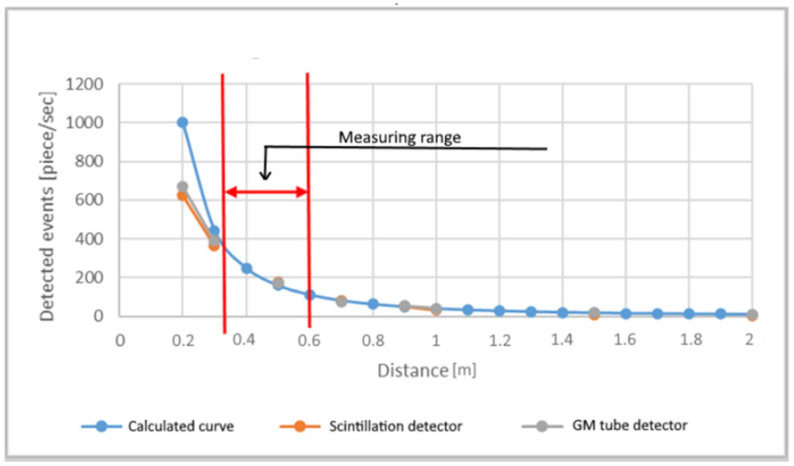
Determining efficient measuring distance.

**Figure 5 sensors-21-04930-f005:**
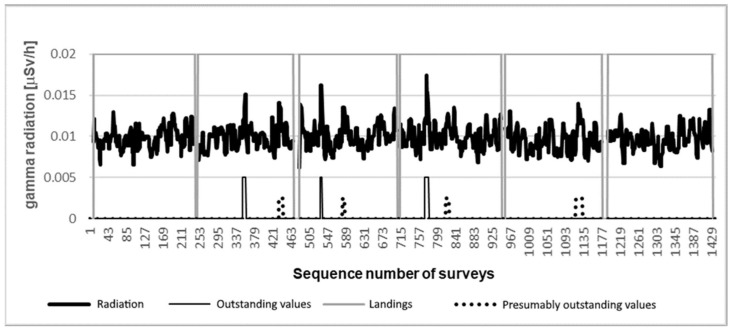
The median filtered data of a radiation measurement over an area of 650 m^2^ and the interpretation of given peaks.

**Figure 6 sensors-21-04930-f006:**
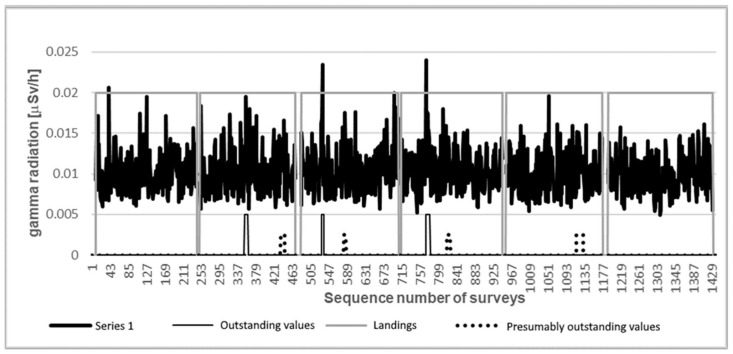
Unfiltered radiation data from the experimental measurement.

**Figure 7 sensors-21-04930-f007:**
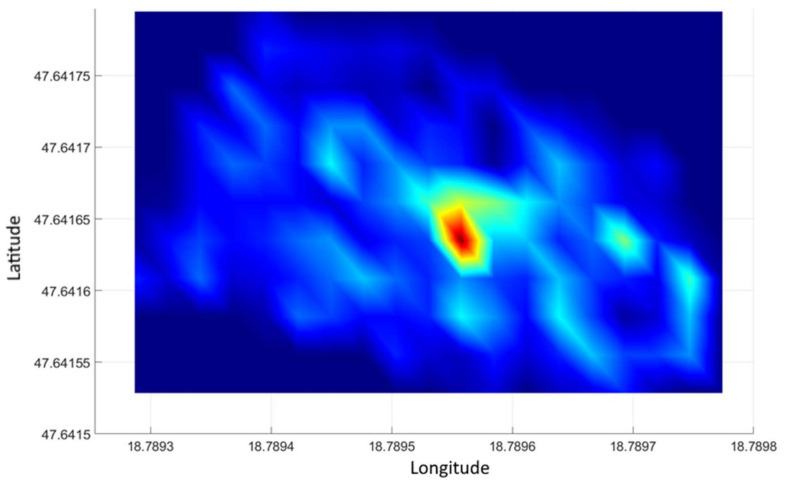
Distribution of gamma radiation intensity in the experimental area with 3 × 3 m processing units.

**Figure 8 sensors-21-04930-f008:**
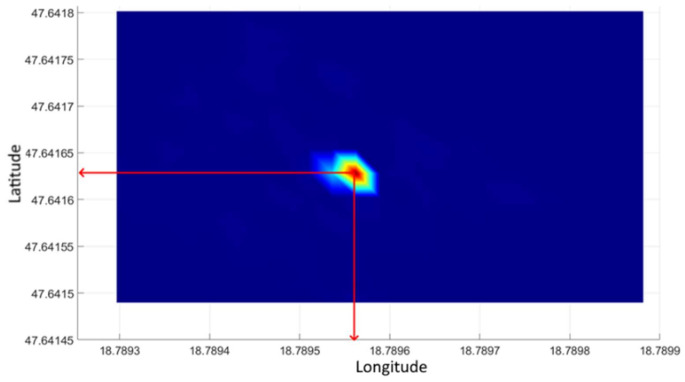
Measurement of gamma radiation controlling the test area using 3 × 3 m processing units.

**Figure 9 sensors-21-04930-f009:**
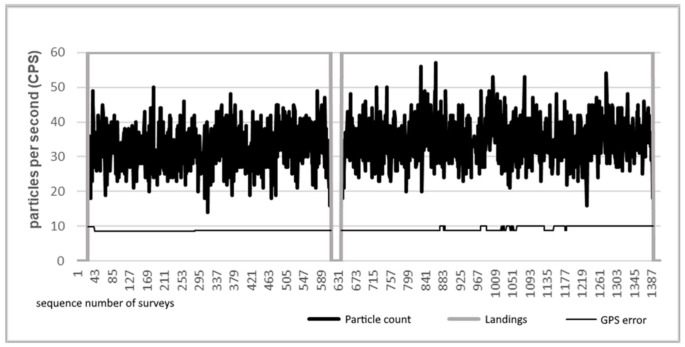
Measurement data of the survey of the testing area: a diagram of the number of events per second during a radiation measurement over an area of 650 m^2^.

**Figure 10 sensors-21-04930-f010:**
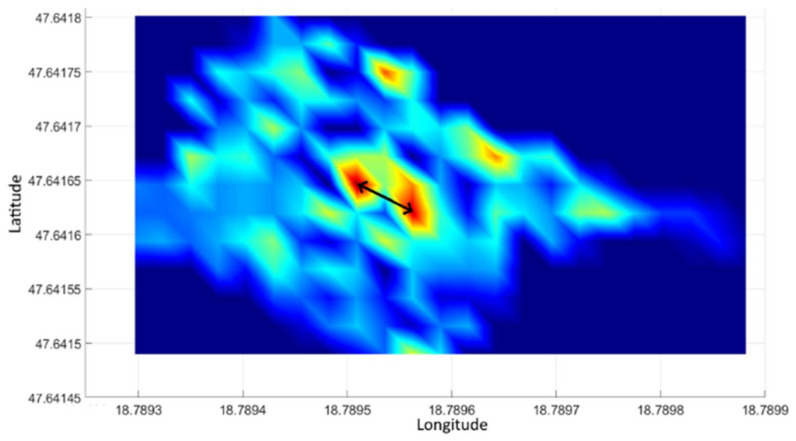
Distribution of gamma radiation intensity in the test area using the parameters of previous processing.

**Figure 11 sensors-21-04930-f011:**
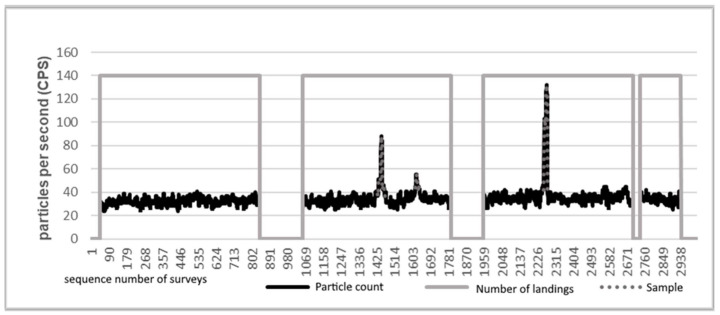
Native measurement data. Display of raw data from the dual-source survey.

**Figure 12 sensors-21-04930-f012:**
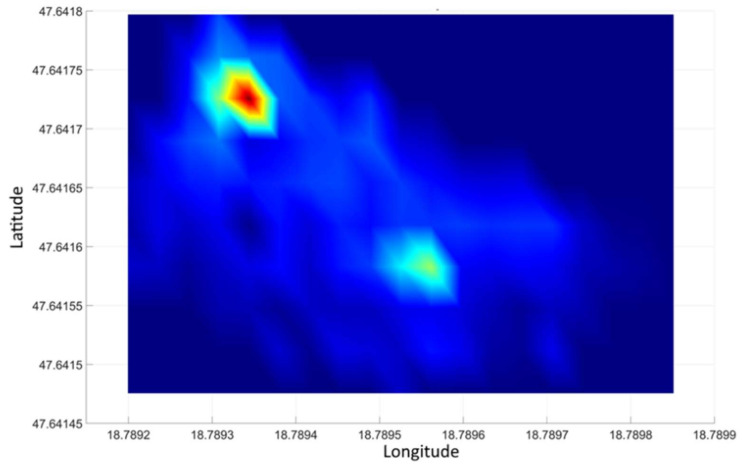
Visual representation of the data from dual-source surveys.

**Figure 13 sensors-21-04930-f013:**
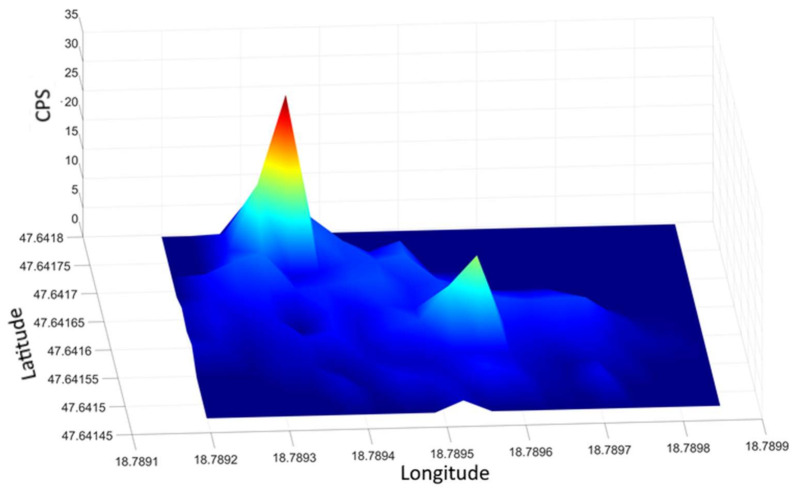
3D display of the data from the dual-source survey, where the altitude is proportional to the count per second (CPS). In the figure, the data were corrected using the average value of the background radiation, thus there are extra detection values along the “Z” axis compared to the background.

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
