# Peer review of "Drone-Based Gamma Radiation Dose Distribution Survey with a Discrete Measurement Point Procedure"

_sensors, 2021, doi:10.3390/s21144930_

Round 1

Reviewer 1 Report

Review

The authors presented an article entitled: Drone-based gamma radiation dose distribution survey with a  discrete measurement point procedure. This manuscript present a certain concept of the authors. I am a bit lacking an explanation of why, in what situations it will be a much better effect, why did the authors take care of it, what better result will we get as a result of choosing their method? We (scientists) improve some solutions or introduce them for a specific purpose. I have not found this goal. It would be good if the authors, apart from the technical aspects of the proposed solution, discussed the issue of its superiority over others. Here I also refer to the lack of discussion of the state of knowledge on this topic in the introduction to the article.

Also, please correct the article as an editor. You can find specific indications below.

Abstract

Line 14 and 15 are the same: two the same sentences  in “abstract”

Line 15 and 17 the same situation: authors repeat the same sentences or fragments of sentences. The content of the abstract should be edited.

Introduction: In the introduction, the authors should present the current state of knowledge on the topic presented. Then present your contribution to the topic discussed.

  1. Practical experiences

equations: The authors should to work on equations. In some of them the components of the equation are not explained (“k” in line 160 and 162), or a different font (mark) was used in the pattern and description, e.g. line: 219 and 222 (I0  and I0) or in equation is “µ “ and “m” in line 223 (m: coefficient of the weakening of the medium)?

Line 231: r2 or r2 ?

Two sentences of comments on Figure 13, please

Other comments:

GIS, CPS, DC – please to explain these abbreviations

Author Response

Dear Sir/Madam,

We corrected the mentioned parts of our article and abbrevations, indexes etc. like:
-Line 14 and 15 are the same: two the same sentences  in “abstract”
-Line 15 and 17 the same situation: authors repeat the same sentences or fragments of sentences. The content of the abstract should be edited.
-Line 231: r2 or r2
-Practical experiences
equations: The authors should to work on equations. In some of them the components of the equation are not explained (“k” in line 160 and 162), or a different font (mark) was used in the pattern and description, e.g. line: 219 and 222 (I0  and I0) or in equation is “µ “ and “m” in line 223 (m: coefficient of the weakening of the medium)?
- GIS, CPS, DC – please to explain these abbreviations

We corrected the introduction section as you asked.
"Introduction: In the introduction, the authors should present the current state of knowledge on the topic presented. Then present your contribution to the topic discussed."
--Our answer which is also placed now in our article:
The main advantage of the system developed and introduced by us over the survey procedures used in practice is that it is easy to mobilize, a large area can be surveyed at low cost without putting an operator at risk in the field. The purpose of the system is to detect the presence of the source and to localize it to such an extent that the localization can subsequently be easily specified by manual or other ground procedures. Since we do not strive for centimeter positioning accuracy, standard GPS localization is sufficient for measurements. During the measurements, the geographical coordinates are interpreted in the WGS’84 system. The circles of latitude and longitude coordinates are also shown in this system in the figures presented.

Reviewer 2 Report

Dear Authors,

Interesting article showing the wide application of UAVs to detect radiation hazards. The presented research and analysis results require improvement. Due to the lack of references in spatial studies, I recommend UAV tests equipped with the GNSS RTK system. The special flight control software described in the introduction is not described in this article. I have provided detailed notes on the manuscript below:

Line 14-17: Repetitions of content. Please correct it.

Line 199: Autinite, Error - Correction.

Line 222, 223: Errors in the marking of I0 and m instead of μ

Line 230: subscript (AD)

Figure 4. Distance should be in [m] not [dm].

Figure 7,8,10, 12 and 13: No axis description in the charts and no definition of the coordinate system (EPSG). The description of the reference system (geographical coordinates) is only in line 300 after Fig. 8. There is also no color scale indicating the level of detected radiation and information about the distance scale. What interpolation methods were used for the spatial distribution of Gamma radiation imagery?

Line 309: There are no arrows on the Fig. 7 and 8. Please clarify what does that mean.

Please explain what GPS error means in Fig. 9 and enter appropriate markings or descriptions in the drawing. The text only explains that GPS Accuracy is about a decrease in the number of satellites whose signals are being tracked by the drone? DJI Mavic series drones already use two global positioning systems GPS and Glonass in navigation, so it is good to use the term GNSS (Global Navigation Satelite System). If the drone model did not use the real-time GNSS (RTK) method for positioning, the accuracy of determining the position was actually not high. In this type of research, it would be beneficial to test an aircraft equipped with GNSS RTK.

Line 359: Again, the arrows pointing to two source points are mentioned. Please complete the drawing.

Figure 13 .: Describe the coordinate axes.

How were the coordinates of the radioactive sources for detection determined and with what accuracy?

There is no information on how accurate the identification of the source locations was for the individual detection methods and air platforms used. Such an analysis would be intentional.

Author Response

Dear Sir/Madam,

We corrected the mentioned parts of our article and abbrevations, indexes, figure errors etc. like:
-Line 14-17: Repetitions of content. Please correct it.
-Line 199: Autinite, Error - Correction.
-Line 222, 223: Errors in the marking of I0 and m instead of μ 
-Line 230: subscript (AD)
-Figure 4. Distance should be in [m] not [dm].
-Figure 7,8,10, 12 and 13: No axis description in the charts and no definition of the coordinate system (EPSG). The description of the reference system (geographical coordinates) is only in line 300 after Fig. 8.
-Line 309: There are no arrows on the Fig. 7 and 8. Please clarify what does that mean.
-Line 359: Again, the arrows pointing to two source points are mentioned. Please complete the drawing.
-Figure 13 .: Describe the coordinate axes.

Reviewer's question:
Due to the lack of references in spatial studies, I recommend UAV tests equipped with the GNSS RTK system. The special flight control software described in the introduction is not described in this article.
--Our answer which is also placed now in our article:
For this phase of the survey, we developed a flight control software solution. Unfortunately, the flight control software available for DJI aircraft fundamentally supports aerial photography, thus hovering the drone over a position for a long time is not possible. The software we developed runs under the Android operating system and was created using the SDK provided by DJI. The software is essentially a route planner which allows hovering at user-set heights for evenly spaced waiting points along the route. Thus, it was possible to reliably repeat the surveys even at extremely low altitudes (2-3m above ground level). The hovering time assigned to the waiting points can also be set by the user. The software checks the planned flight program and compares it with the maximum flight time with the battery capacity of the given DJI aircraft. If the plan involves a flight longer than what the drone is capable of, it will not allow takeoff. In this case, several flight plans must be prepared which cover smaller sections and they must be flown one after the other. 

Reviewer's question:
Please explain what GPS error means in Fig. 9 and enter appropriate markings or descriptions in the drawing. The text only explains that GPS Accuracy is about a decrease in the number of satellites whose signals are being tracked by the drone? DJI Mavic series drones already use two global positioning systems GPS and Glonass in navigation, so it is good to use the term GNSS (Global Navigation Satelite System). If the drone model did not use the real-time GNSS (RTK) method for positioning, the accuracy of determining the position was actually not high. In this type of research, it would be beneficial to test an aircraft equipped with GNSS RTK.
--Our answer to the questions above which are also placed now in our article:
Given that our goal was to implement a small, easy-to-transport system, one of the carrier systems was a DJI Mavic Pro multi-copter. Due to the limited load capacity, the required minimum has been determined for onboard systems. Of course, in the case of RTK localization, determining the measurement points would provide an accuracy of less than a centimeter, but we did not aim for this accuracy in our system. According to our research and development concept, the measurement accuracy of the standard GPS (for civilian use) was sufficient. 

Reviewer's question:
How were the coordinates of the radioactive sources for detection determined and with what accuracy?
--Our answer which is also placed now in our article:
The purpose of the system is to detect the presence of the source and to localize it to such an extent that the localization can subsequently be easily specified by manual or other ground procedures. Since we do not strive for centimeter positioning accuracy, standard GPS localization is sufficient for measurements. 

Reviewer's question:
There is no information on how accurate the identification of the source locations was for the individual detection methods and air platforms used. Such an analysis would be intentional.
--Our detailed answer is equal our answer to your previous question:
Given that our goal was to implement a small, easy-to-transport system, one of the carrier systems was a DJI Mavic Pro multi-copter. Due to the limited load capacity, the required minimum has been determined for onboard systems. Of course, in the case of RTK localization, determining the measurement points would provide an accuracy of less than a centimeter, but we did not aim for this accuracy in our system. According to our research and development concept, the measurement accuracy of the standard GPS (for civilian use) was sufficient. 

Reviewer 3 Report

This paper presents the results of two experiments based on Geiger-Muller and scintillation detectors onboard in a drone for measuring gamma radiation from a uranium mineral positioned in the field. Because the manuscript does not follow traditional structure (Introduction/Materials and Methods/Results & Discussion/Conclusions), I had hard time to understand the scope of the paper. Other comments are:

1. In the Introduction section, authors should provide the state-of-the-art of the subject based on literature review and then state what is the main contribution of the paper. It is expected that the most important papers in the field of study are cited in this section. The last paragraph should be the objective of the study. Instead, the objective is stated in the first paragraph (the primary objective of the development was to create a more compact, easily portable, and deployable system, but one which, in contrast with the previous ones, is more sensitive). I can also find results in this section which is also not appropriated (in this present development, this value was successfully reduced to +0.005 - +0.007 mS/h; the improvement in sensitivity was achieved primarily by increasing the measurement time per point, which was realized using special flight control software).

2. In Section 2 (Applied detectors), authors should state the existing detectors to measure gamma radiation (at least the most important ones) and the reasons for choosing Geiger-Muller and scintillation detectors for this study.

3. The meaning of X-axis and Y-axis is unclear in Figure 2.

4. Authors present the results & discussion without citations. In this way, the manuscript seems to be more likely a preliminary report of a research than a scientific paper.

5. The number of citations is quite few (14, about half from scientific journals).

Author Response

Dear Sir/Madam,

We corrected the mentioned parts of our article.

Reviewer's question:
1. In the Introduction section, authors should provide the state-of-the-art of the subject based on literature review and then state what is the main contribution of the paper. It is expected that the most important papers in the field of study are cited in this section. The last paragraph should be the objective of the study. Instead, the objective is stated in the first paragraph (the primary objective of the development was to create a more compact, easily portable, and deployable system, but one which, in contrast with the previous ones, is more sensitive). I can also find results in this section which is also not appropriated (in this present development, this value was successfully reduced to +0.005 - +0.007 mS/h; the improvement in sensitivity was achieved primarily by increasing the measurement time per point, which was realized using special flight control software).
--Our first answer:
We put more citation to the important papers.
--Our answer which is also findable in our original article in the second chapter:
From radiation-sensitive detectors which can be mounted to a drone, two were chosen for testing.

Reviewer's question:
2. In Section 2 (Applied detectors), authors should state the existing detectors to measure gamma radiation (at least the most important ones) and the reasons for choosing Geiger-Muller and scintillation detectors for this study.
--Our answer which is also placed now in our article:
Given that our goal was to implement a small, easy-to-transport system, one of the carrier systems was a DJI Mavic Pro multi-copter. Due to the limited load capacity, the required minimum has been determined for onboard systems. 
The main advantage of the system developed and introduced by us over the survey procedures used in practice is that it is easy to mobilize, a large area can be surveyed at low cost without putting an operator at risk in the field.

3. The meaning of X-axis and Y-axis is unclear in Figure 2.
-- we corrected with many other abbrevation, figure etc. error.

4. Authors present the results & discussion without citations. In this way, the manuscript seems to be more likely a preliminary report of a research than a scientific paper.
--Our answer:
We cited many important papers in the field of study. Although not cited them again in the results, but if anybody follow citations, it can be recognizable that our system is more lighter, easier to use than others with the same accuracy.

5. The number of citations is quite few (14, about half from scientific journals).
--Our answer:
Yes, we missed some important paper from this field. We corrected now.

Round 2

Reviewer 3 Report

The revised version of the manuscript is much improved now. Most of my concerns were addressed properly. Two minor comments:

  1. There is room to improve the titles of Figures 1 to 4.
  2. Titles inside the Figures 4-6 is unnecessary.
  3. Section 5 (Summary) can be changed to Conclusions, which is the style used by Sensors.

Author Response

Dear Sir/Madam,

Thank you for your second round review. We corrected our paper based on them.